# Quality of Nursing Work Life, Compassion Fatigue, and Self-Efficacy Among Primary Care Nurses in Saudi Arabia

**DOI:** 10.3390/healthcare13151811

**Published:** 2025-07-25

**Authors:** Hind Al-Otaibi, Ali Kerari

**Affiliations:** 1Community and Mental Health Department, College of Nursing, Imam Abdulrahman Bin Faisal University, Dammam 34212, Saudi Arabia; hmkalotaibi@iau.edu.sa; 2Nursing Administration and Education Department, College of Nursing, King Saud University, Riyadh 11421, Saudi Arabia

**Keywords:** quality of life, professional quality, compassion, primary care, nursing

## Abstract

**Background/Objectives**: This quantitative cross-sectional study examined the relationships between self-efficacy, compassion fatigue, and the quality of nursing work life (QNWL) in primary care nurses from the Dammam and Riyadh regions of Saudi Arabia. This study examined how these factors varied according to gender, education, income, and years of experience. **Methods:** A total of 158 primary care nurses completed validated survey instruments, including the General Self-Efficacy Scale, Professional Quality of Life Scale—Version 5, and Work-Related Quality of Life Scale-2. Descriptive statistics were used for data analysis. **Results:** The results indicated that participants experienced elevated self-efficacy (M = 29.53, SD = 0.52), moderate compassion fatigue (M = 54.62, SD = 10.16), and moderate overall quality of work life (QWL) (M = 3.26, SD = 0.52). Positive correlations were identified between self-efficacy and QWL (*r* = 0.250, *p* < 0.05) and compassion satisfaction (*r* = 0.533, *p* < 0.05). By contrast, compassion fatigue was negatively correlated with QWL (*r* = −0.259, *p* < 0.05). Notable disparities in QWL were identified according to education level and years of experience, with nurses possessing advanced degrees and those with less experience having elevated QNWL scores. **Conclusions:** This study highlights the significance of promoting self-efficacy and mitigating compassion fatigue to improve the quality of nursing. Administrators and nurse leaders should consider strategies such as continuous professional development, psychological support, and effective workload management to enhance the well-being and retention of primary care nurses.

## 1. Introduction

The nursing profession is integral in providing high-level medical care to patients. Achieving the best possible patient outcomes depends primarily on nurses with significant competencies and roles [1]. Quality of working life (QWL) refers to the process through which organizations actively enable their employees—regardless of their rank—to shape their work environment, processes, and outcomes [2]. The two goals of this value-based approach are to increase employees’ quality of life (QOL) in the workplace and improve organizational effectiveness. QWL refers to the understanding of health workers, work, and organizations. The main components are belief in participation in organizational decision-making and problem-solving, as well as concern for the impact of work on health workers and organizational effectiveness [2].

Nurses’ quality of professional life is considered a fundamental element for the further development of healthcare. QWL affects job satisfaction, which in turn affects nursing staff performance. Positive outcomes, such as increased quality of care, higher productivity and performance, increased retention rates, and lower turnover, are related to nurses’ satisfaction with their QWL. Therefore, improving the quality of nursing life in the workplace is essential to provide better nursing services. Many factors have been shown to affect the work life of nurses, including work overload, lack of work–life balance, and the work environment [3].

The critical role of nurses in the global healthcare system is undeniable, especially given the high demands and stress that characterize their professional fields. From human resources management and healthcare administration, the importance of emotional well-being, along with self-efficacy and emotional intelligence, has emerged as a primary focus of recent research [4]. Self-efficacy refers to the conviction and assurance of one’s capacity to overcome challenging circumstances or stressful situations [5]. A person’s sense of their ability to perform the behaviors required to achieve a particular outcome is known as self-efficacy, and in this regard, it is essential to shift from knowledge to behavior. Self-efficacy is a key factor determining psychological adjustment when dealing with stressful situations [5]. Self-efficacy affects motivation and confidence in overcoming obstacles and enhances performance, which can impact behavior [6]. Research suggests that employees with high levels of self-efficacy believe that they are capable of managing their patients’ symptoms [7]. Previous studies have also revealed a substantial positive correlation between self-efficacy and organizational climate [7,8]. As a result, nurses who exhibited higher levels of self-efficacy and were more devoted to their organization were also more satisfied with their work environment.

Compassion is recognized as a fundamental professional value by the International Council of Nurses (ICN), as outlined in its Code of Ethics for Nurses [9]. It is often regarded as an essential virtue of exemplary nursing practice, which is described as an intrinsic drive to alleviate human suffering. Compassion includes attributes such as empathy, kindness, patience, and the ability to provide comfort. While organizations may not mandate compassion, it is essential that nurses genuinely experience it, and this quality can be cultivated through education [10,11].

Inadequate compassionate care can occur when nursing is reduced to merely technical procedures, regardless of the proficiency with which these tasks are executed, thereby undermining the ethical and compassionate essence of the profession [10,11]. Nurses may perform physical duties without emotional involvement or considering the patient’s need for emotional support. According to Jean Watson’s Caritas Process, nurses are urged to be fully present in the moment and not merely execute the actions involved in caring for patients [12]. The lack of presence during care moments, when nurses mechanically perform technical tasks without engaging with the patient, can be attributed to emotional stress resulting from prolonged exposure to suffering patients. This leads to compassion fatigue, which encompasses emotional, physical, and spiritual exhaustion [13].

Compassion fatigue is a complex phenomenon that can significantly impact not only nurses but also the patients they care for and the healthcare organizations they serve [14]. It can strike without notice, leaving a person feeling emotionally spent, frustrated, powerless, and without a sense of accomplishment at work, especially among nurses [15]. Elevated levels of compassion fatigue have been associated with poorer care, more unfavorable outcomes, and less caring relationships [16]. In contrast to other healthcare professionals, nurses frequently interact directly with patients and witness their anguish daily. They also face heavy workloads, shift rotations, and unfavorable working conditions [10]. Therefore, they are more susceptible to compassion fatigue than are other healthcare professionals. According to earlier research, up to 52.55% of clinical nurses report compassion fatigue [11]. Thus, nurses’ compassion fatigue has received increasing attention, contributing to efforts to prevent related unfavorable consequences.

Moreover, the quality of nurses’ work–life balance and their stress levels are important to consider [17]. The interplay between self-efficacy, compassion fatigue, and the quality of nursing work life (QNWL) is critical, as higher self-efficacy can buffer against compassion fatigue and enhance QNWL, while excessive compassion fatigue may negatively affect both self-efficacy and nurses’ perceptions of their work environment. Understanding these interconnections provides a comprehensive perspective on how to support nurses’ well-being and improve patient care outcomes. However, a gap in knowledge remains regarding the interrelationships between self-efficacy, compassion fatigue, and QNWL.

In Saudi Arabia, nurses deliver care within a distinctive cultural and social framework that shapes their professional experiences and well-being. Expat and Saudi nurses often navigate cultural expectations related to family obligations and community perceptions of nursing as a profession. For example, many Saudi nurses balance traditional family expectations with professional demands, and community views on nursing may influence job responsibilities and career development. Primary care nurses play a crucial role in health promotion and disease prevention across diverse communities that are rapidly evolving under Saudi Arabia’s Vision 2030. These contextual factors may influence levels of compassion fatigue, compassion satisfaction, and self-efficacy, underscoring the need to better understand the working conditions of nurses in the Saudi healthcare system. Recent studies have shown that Saudi nurses continue to face challenges in their quality of work life, highlighting the need for updated evidence and practical interventions [1,3]. Primary care nurses’ well-being is therefore critical for achieving Saudi Arabia’s Vision 2030 healthcare goals.

In the context of Saudi Arabia, there are limited data regarding the quality of nurses’ working lives, despite their critical role in primary care settings, where they play a key role in the prevention and health promotion of individuals at all stages of life. Additionally, more studies are needed to address fundamental concerns regarding professional QOL (i.e., compassion fatigue and satisfaction) to ascertain the determinants of the quality of professional life experienced by primary care nurses. Therefore, the objective of this study was to investigate the relationships between self-efficacy, compassion fatigue, and QNWL among primary care nurses in Saudi Arabia. Another aim was to analyze how these variables differed based on individuals’ demographic characteristics, providing insights to support strategies that can enhance nurses’ well-being and patient care quality. The following three research questions guided this study.
What are the descriptive statistics of the variables QNWL, self-efficacy, compassion fatigue, and compassion satisfaction among primary care nurses?What are the relationships between QNWL, self-efficacy, compassion fatigue, and compassion satisfaction among primary care nurses?Are there differences in QNWL, self-efficacy, compassion satisfaction, and compassion fatigue according to the nurses’ demographic characteristics (age, sex, education, income, work experience)?


## 2. Materials and Methods

### 2.1. Study Design

The study was conducted between July and November 2024. The study employed a descriptive cross-sectional design, which allowed the analysis of the relationship between QNWL, self-efficacy, and compassion (fatigue and satisfaction). This design was selected based on its practicality and ability to fulfill the study objectives within a limited timeframe.

### 2.2. Sample

The target population was nurses working in Saudi primary healthcare centers (PHCs), and the sample frame included primary healthcare nurses from several PHCs in Riyadh, Dammam, and Khobar. We recruited a purposive sample of primary care nurses. The inclusion criteria were as follows: those with more than one year of experience in PHCs and who could understand and read English. Nurses with less than one year of experience in PHCs were excluded.

The sample size was identified using statistical power analysis. Significance criterion (α), population effect size (size of the correlation of the independent variables to the dependent variables), and statistical power are important parameters included in statistical inference [18]. With regard to sample size calculations, the variable numbers, covariates, and statistical tests to be performed were determined. Thus, G-power software was used to calculate the required sample size, with the significance level set at α = 0.05. The population effect size was estimated to be f2 = 0.15, representing the anticipated strength of the association between the independent and dependent variables [18]. Based on the guidelines for power analysis, the appropriate sample size was estimated to be 114. The final sample size increased to 136 participants, accounting for 20% attrition. However, a total of 158 individuals participated, exceeding the calculated sample size and thereby increasing the robustness and generalizability of the findings. The sample included nurses of various ages and genders, with the majority being female and aged between 31 and 45 years, which aligns with the typical demographics of primary care nurses in Saudi Arabia.

### 2.3. Data Collection Procedure

The study questionnaire was an online survey, and data were collected using online platforms that allowed participants to complete the form on devices such as smartphones and tablets. This approach is efficient and cost-effective. All necessary clearances were acquired, and a research team member reached out to the administration of PHCs in several Saudi Arabian regions, providing a detailed explanation of the study and requesting permission for the recruitment of participants who met the inclusion criteria. Questions were answered online. The researcher developed a Google Forms questionnaire to facilitate the online collection of quantitative data. The first page of the online questionnaire included an informed consent statement, followed by demographic questions and quantitative assessment tools. The final responses were extracted using Microsoft Excel.

### 2.4. Measurements

The survey was conducted in English, which is widely recognized as a professional language by healthcare professionals in Saudi Arabia. In addition to demographic information (age, sex, income, marital status, educational level, years of experience, and overall health status), this study used three instruments in an online multiple-section questionnaire.

### 2.5. Work-Related Quality of Life Scale-2

The Work-Related Quality of Life Scale-2 (WRQoL-2) is a psychometric tool that utilizes seven psychosocial factors to assess QWL, providing researchers, organizations, and individuals with insights into employee well-being and the impact of work on QOL [19]. It consists of 32 items, with one question representing the perception of total QWL excluded from scoring. The remaining 31 variables were categorized into seven factors: control at work (CAW), employee engagement (EEG), general well-being (GWB), home–work interface (HWI), job career satisfaction (JCS), stress at work (SAW), and working conditions (WCS). Each item was evaluated using a 5-point Likert scale, with one representing “strongly disagree” and five indicating “strongly agree” [19]. In this study, Cronbach’s alpha was 0.91.

### 2.6. Professional Quality of Life Scale—Version 5

The Professional Quality of Life Scale—Version 5 (ProQOL V5) includes 30 Likert-type items used to evaluate compassion satisfaction (CS) and compassion fatigue (CF). The ProQOL V5 consists of three scales: CS, burnout, and secondary traumatic stress (STS). The participants were asked to rate their interactions with their patients over the past 30 days [20]. Participants were instructed to complete the survey by considering their current practice and respond to each question using a 5-point Likert scale (1 = never, 2 = rarely, 3 = sometimes, 4 = often, 5 = very often), reflecting the frequency of feelings described in the items over the past 30 days. The scores for each subscale were combined to obtain a total raw score; the minimum score for each subscale was 10, and the maximum score was 50; the raw scores can be categorized as low, medium, or high [20].

In Saudi Arabia, Bahari et al. employed the ProQOL V5 among hospital nurses in Saudi Arabia and reported satisfactory internal consistency, with Cronbach’s alpha values exceeding 0.70 across the three subscales [21]. Similarly, Alabd et al. used the scale with primary healthcare nurses in Madinah and confirmed its cultural relevance and psychometric robustness [22]. These findings support the reliability and validity of the scale for use in Saudi healthcare settings, particularly among nursing professionals. In this study, Cronbach’s alphas for the three subscales were 0.91, 0.71, and 0.84, respectively.

### 2.7. The General Self-Efficacy Scale

The General Self-Efficacy Scale (GSE) was developed in German by Schwarzer et al. [23]. The authors aimed to evaluate individuals’ overall perception of self-efficacy, predicting their ability to cope with daily pressures and adapt to stressful events [23]. The average time taken to complete the GSE is 5 min. The response options are based on a four-point Likert scale, ranging from not at all true = 1 to exactly true = 4. All 10 items provide a final composite score between 10 and 40, with elevated values indicating more generalized self-efficacy. The items on the scale indicate an individual’s positive assessment of personal achievement. An example item is, “I can always manage to solve difficult problems if I try hard enough” [23]. In this study, Cronbach’s alpha was 0.90.

### 2.8. Ethical Considerations

The Ethics Committee of King Saud University approved this study. Before commencing, the proposal was submitted to the Institutional Review Board (IRB) of King Saud University for ethical approval. Upon approval, the researcher informed the nursing directors of the PHCs and obtained institutional support to conduct the study.

Eligible participants were informed that their participation was voluntary and they had the right to withdraw at any time without penalty. They were also allowed to ask questions throughout the study period. Anonymity and confidentiality were strictly maintained, and all collected data were securely stored for a specified period before destruction. Access to the data was restricted to the researcher and IRB personnel if required.

Participants were assured that there were no anticipated risks associated with their participation. Although no direct benefits were observed, the findings of this study could contribute to improving the quality of patient care. To ensure confidentiality, all the questionnaires were completed anonymously.

### 2.9. Data Analysis

All statistical analyses were performed using SPSS Version 29. Descriptive statistics (frequencies, percentages, and means) were used to describe sample variables. Parametric independent sample t-tests and one-way analysis of variance (ANOVA) were conducted to assess significant differences between the groups. Pearson’s correlation coefficient was used to examine the relationships among variables. All assumptions related to statistical tests were addressed. Statistical significance was set at *p* < 0.05.

## 3. Results

### 3.1. Demographic Characteristics of the Participants

From the initial sample, nine respondents refused to participate in the online survey, and of the remaining 164 eligible participants, six were excluded owing to missing data or incomplete submissions. The total number of primary care nurses who completed the survey was 158, with a response rate of 92%. Male nurses comprised 31.6% (*n* = 50) of the participants, and female nurses accounted for 68.4% (*n* = 108). Most were aged between 31 and 45 years (*n* = 63.9%), while others were older than 46 years (11.4%). Participants’ educational levels were categorized as follows: (1) diploma degree in nursing, accounting for 47, or 29.7% of the total participants; (2) bachelor of science in nursing (BSN), accounting for 68, or 43% of all participants; and (3) master’s degree or higher in nursing, accounting for 43, or 27.2% of the total participants. Most primary care nurses (63.3%) reported 10 or more years of work-related experience, and 36.7% of the participants had less than 10 years of work-related experience. Most participants had higher income levels, representing 69% (*n* = 109) of the study sample, as shown in Table 1.

### 3.2. General Self-Efficacy

The mean general self-efficacy score was 3.69 (SD = ±0.74), indicating that many participants experienced moderate levels of self-efficacy, as reported by 51.8% of participants (see Table 2). The independent t-test and one-way ANOVA results showed no significant differences between the main study variables and individual characteristics (*p* > 0.05).

### 3.3. WRQoL-2

The results showed that the overall WRQoL-2 mean score was 3.26 (±0.52), indicating that the QWL of primary care nurses was moderate. The results also revealed that the highest average score for JCS was 3.65 (±0.67), and the lowest mean score was SAW 2.46 (±0.52) (see Table 3). The mean scores of other WRQoL factors were as follows: CAW, 3.50 ± 0.79; EEG, 3.55 ± 0.85; GWB, 3.51 ± 0.60; HWI, 3.50 ± 0.79; and WCS, 2.60 ± 0.61. As seen in Table 4, a significant difference was found between work-related experience levels and QWL in primary care settings (t (156) = 2.28, *p* = 0.02, %95 CI [0.18, 2.54]). Participants with fewer years of work experience had higher work-related QOL. Further, there was a significant difference between QWL and educational level (*p* < 0.05). Participants with an MSN or higher reported higher mean scores on the WRQoL-2 than those with diplomas (f (155) = 4.48, *p* = 0.01, %95 CI [0.41, 4.09]). There were no significant differences in QWL based on gender and income (*p* > 0.05) (see Table 4).

### 3.4. Subscales of Professional QOL (Compassion Satisfaction and Compassion Fatigue)

In this study, the overall CS subscale mean was 3.58 (SD = 0.73), indicating that primary care nurses were highly satisfied with their work (see Table 2). The mean total score on the burnout subscale was 2.70 (SD = 0.44), indicating that the participants were at a moderate risk of burnout. The mean of the overall scores on the CF subscale was 2.73 (SD = 0.50), indicating that the participants perceived a moderate feeling of CF. The independent t-test results showed that participants with higher income levels reported greater CS with their work (t (156) = 2.24, *p* = 0.02, %95 CI [0.33, 5.27]). Additionally, there was a significant difference between CF and educational level. Participants who obtained diplomas perceived higher levels of CF than those with MSN or higher degrees in nursing (f (155) = 5.07, *p* = 0.003, %95 CI [1.99, 12.11]). (see Table 4). No significant differences were found between other individual characteristics (i.e., gender and work experience) and CF or satisfaction.

### 3.5. Associations Between QNWL, Self-Efficacy, CF, and CS

The relationships between QWL, self-efficacy, CF, and CS are presented in Table 5. A positive correlation was noted between self-efficacy and the QWL variables (*r* = 0.250; *p* < 0.001), suggesting that individuals with elevated self-efficacy were likely to report enhanced QWL.

CS was positively correlated with QWL (*r* = 0.233, *p* < 0.001), indicating that obtaining fulfilment from compassionate care improved the overall workplace experience. Conversely, CF demonstrated a negative correlation with QWL (*r* = −0.259, *p* < 0.001). In addition, the correlation between self-efficacy and CS was significantly robust (*r* = 0.533, *p* < 0.001), indicating that individuals who perceived themselves as competent in their roles were more inclined to experience satisfaction from compassionate endeavors.

## 4. Discussion

This quantitative study investigated the relationships between QWL, CF, CS, and self-efficacy among primary care nurses in Saudi Arabia. The findings, summarized below, could be valuable to improve nurses’ well-being, the quality of healthcare services, and patient care.

### 4.1. General Self-Efficacy

The results showed that many of the participants experienced moderate levels of self-efficacy, as reported by 51.8%. Our results, in line with Andargeery et al.’s findings, indicate that nursing students possess a moderate perception of their self-efficacy [24]. These findings are consistent with those of previous studies [25,26], which revealed that nursing professionals perceived their self-efficacy and self-esteem as moderate to high. Healthcare providers, who are capable of performing tasks to a degree comparable to most individuals, possess a range of positive attributes and skills, have high self-assessments of their creative abilities, maintain an optimistic outlook on life, and believe that their value is equal to that of others.

Our results revealed a statistically significant relationship between self-efficacy and compassion satisfaction. This strong positive correlation indicates that individuals with higher self-efficacy or confidence in their ability to handle challenges tend to experience greater compassion satisfaction acts. This is expected in such work environments, especially in healthcare, social work, and education, where helping others is at the core of related roles. Self-efficacy could potentially buffer against emotional exhaustion and enhance satisfaction through the provision of care or assistance.

Establishing ongoing training and educational initiatives is a vital administrative approach for enhancing the self-efficacy of nursing personnel. By equipping nurses with essential skills and resources, they can gain confidence in their everyday tasks while advancing their career growth [27]. Leaders must cultivate an atmosphere that offers successful experiences, constructive feedback, and positive role models. In environments where nurses’ efforts are acknowledged and appreciated, they tend to experience greater self-efficacy, which positively affects their work engagement. Ultimately, investing in enhancing nurses’ self-efficacy benefits the individuals and significantly elevates the quality of healthcare services and patient satisfaction [21,28,29].

Our findings reveal a noteworthy, albeit modest, relationship between self-efficacy and QOL at work. This aligns with Orgambídez’s findings [8]. This connection suggests that self-efficacy contributes to a favorable work experience, potentially empowering individuals to handle job demands and improving their QOL in the workplace. The notable positive connection between self-efficacy and CS identified in this study indicates that individuals with higher levels of self-efficacy derive more satisfaction from their compassionate roles.

This link suggests that confident employees may find assisting others more fulfilling as they perceive themselves as competent and effective in their caregiving tasks. The significant associations between self-efficacy, work-related QOL, CS, and overall health status indicate that these elements contribute positively to workplace experience and employee well-being. Self-efficacy plays an essential role in how nurses confront and navigate challenges in work settings. From an administrative perspective, these results highlighted the importance of creating an environment that nurtures and supports self-efficacy. When nurses feel competent and equipped to tackle workplace challenges, their commitment to the institution increases, enhancing their efficiency and job satisfaction.

Acknowledging and tackling the difficulties that nurses encounter in the global healthcare landscape, particularly in situations characterized by high demand and stress, is crucial. Managing professionals’ self-efficacy, job satisfaction, and overall QOL is an ethical obligation and strategic necessity to improve performance and retain staff [4].

### 4.2. CF and Satisfaction

Our findings revealed a moderate level of CF, aligning with Kim’s study that reported moderate CF levels among nurses [30]. A study in Turkey also indicated that nurses experienced moderate CF levels [31]. On the contrary, Aslan’s study revealed that nurses had an average score of 14.3 ± 7.7 on the CF scale, indicating low levels of CF [32].

The average score for the total burnout subscale was 27 (SD = 4.92), with the lowest mean score for work-related stress being 2.46 (±0.52). CF is a concept described in analyses of prolonged stress exposure. Nurses with a higher frequency of stress are more likely to have increased CF [32].

In this study, primary care nurses experienced a moderate level of CS in their roles. As Aslan noted, the life attitude profile emerged as the most significant factor influencing CF, exhibiting a negative correlation [32]. Life satisfaction plays a crucial role in ensuring effective functioning within society. The recent advent of positive psychology has led to an increase in research exploring aspects that can enhance life satisfaction and foster positive attributes. From this perspective, while chronic emotional distress can be detrimental, various opposing emotions can potentially act as remedies. Compassion has emerged as a prominent concept in the field [33,34]. A noteworthy negative correlation was found between work-related QOL and compassion fatigue. This indicates that improvements in work-related QOL are associated with reduced compassion fatigue. This implies that fostering a supportive work environment, which may include manageable caseloads, regular breaks, and sufficient supervision, can alleviate the emotional burden on care-oriented professionals.

Meanwhile, CS shows a positive correlation with QoWL. This suggests that employees who find fulfillment in their compassionate roles tend to experience higher overall QWL. CS may serve as a protective element, fostering emotional resilience and allowing employees to perceive their work more favorably, even in the face of difficulties. This could be attributed to reduced stress levels and enhanced morale among those who find joy assisting others [10]. Encouraging self-compassion and facilitating compassionate care within the primary healthcare team may enhance both the patient care experience and positive engagement and satisfaction of healthcare professionals.

The negative relationship between CF and work-related QOL indicates how the emotional strain and stress tied to caregiving roles can diminish one’s perception of QWL. Fatigue frequently arises from ongoing exposure to the emotional demands associated with caring for others, especially in high-pressure fields such as healthcare, counseling, and social work. Hendriati and Achmat found that burnout affected the QOL of nurses involved in disaster emergency response [35]. The outcomes of their research are consistent with earlier findings by Lungulescu et al., who showed that the QOL of health workers dealing with COVID-19 in Romania was significantly affected by burnout [36]. Additionally, lower levels of burnout were associated with factors such as personal identity (including satisfaction or contentment with professional responsibilities), self-efficacy, organizational support, and both social and professional identification. These elements contribute to enhanced QOL [37]. In the QOL category, engaging in disaster relief volunteering is strongly associated with better social interactions, physical fitness, and environmental support [38]. Burnout increases employees’ chances of experiencing poor physical and mental health [39]. Prior research has indicated that healthcare professionals characterized by elevated burnout levels also have lower QOL scores and face an increased risk of infections during their work [26,40]. Additional studies have identified factors contributing to volunteer burnout, including high work demands, ambiguity regarding various roles or tasks, grievances during service delivery, lack of organization, prolonged involvement, delays in task schedule adjustments, and stress encountered at disaster sites [41].

Conversely, CF arises when nurses develop close relationships with patients, resulting in the blurring of emotional boundaries that leads them to internalize the suffering of patients, which in turn can evoke feelings of guilt, hopelessness, or powerlessness [42]. Professional competence may help prevent the onset and escalation of burnout and CF because of its various beneficial characteristics, including communication skills, strong interpersonal relationships, effective teamwork, leadership, and ethical decision making. Professional competence can leverage effective communication and leadership skills to foster a constructive work atmosphere and nurture good teamwork and relationships [43]. Furthermore, competent nurses can apply suitable communication techniques, moral principles, and ethical standards to express compassion and empathy in their actions without becoming intertwined with patient suffering, which may result in countertransference, burnout, and CF [42].

### 4.3. QNWL

The results of our study indicated that the QWL of primary care nurses was moderate. Our findings are consistent with those of a previous study involving nurses in Jordan that demonstrated that participants experienced moderate QWL [44]. Conversely, a study conducted in Saudi Arabia revealed that most nurses reported poor QWL [45]. Moreover, Alzoubi found that a significant proportion of nurses (31.2%) described their QWL as fair, while only a small percentage (4.4%) rated it as high [29]. Differences in work environments across academic and clinical settings may have contributed to these conflicting results.

Data analysis showed that sociodemographic factors are associated with QWL. Regarding education level, there was a notable difference in QWL across different educational backgrounds (*p* < 0.05). Participants with MSN or higher degrees had higher average WRQoL-2 scores than those with diplomas or BSN degrees. This aligns with the findings of Alabad et al., who indicated that nurses with bachelor’s degrees and substantial education tended to score higher on QWL [22]. This observation is supported by previous research conducted in Bangladesh and Ethiopia, which identified a statistically significant connection between nurses’ QWL and educational achievement [46,47]. Similarly, Raeissi et al. noted that nurses with advanced education enjoyed higher QWL than those with lower educational qualifications [48].

Furthermore, a statistically significant difference was identified between the levels of work experience concerning QWL in primary care environments (*p* < 0.05). Participants with fewer years of work experience demonstrated elevated levels of work-related QOL. This contradicts the findings of Alabd’s study, which indicated a significant correlation between QWL and work experience, showing that nurses with more years of experience had better QWL [22]. Healthcare providers with less experience may bring fresh enthusiasm and new ideas, contributing to a more positive work environment.

The positive relationships between work-related QOL, compassion satisfaction, and self-efficacy indicate that how one perceives workplace quality directly influences their general satisfaction with their working life. This implies that the elements that enhance job satisfaction, such as sufficient resources, equitable treatment, and impactful work, can enhance an individual’s overall work experience. QWL may play a crucial role in shaping clinical nursing settings to meet the needs and expectations of nurses. Researchers need to assess the real-world environments in which nurses operate and understand the effects of the workplace on both nurses and patients. Enhancing the quality of the work environment, fostering fair policies, and ensuring adequate support may lead to greater overall satisfaction with work life, which is likely to positively influence retention and productivity.

### 4.4. Implications for Nursing Practice and Research

Healthcare organizations should prioritize strategies to strengthen nurses’ professional confidence. Consistent education and training programs, mentorship initiatives, and psychological assistance enhance nurses’ resilience, reduce compassion fatigue, and increase job satisfaction. Moreover, integrating self-efficacy enhancement programs into nursing curricula and continuing education can strengthen nurses’ abilities to address workplace challenges.

The sustainability of both workers and healthcare standards is at risk for CF. Policymakers should create institutional standards that require mental health support services, such as fostering an environment that offers impactful experiences, constructive criticism, and exemplary role models for integration into healthcare organizations. To ensure the effectiveness of the primary care nursing workforce and elevate the quality of healthcare services and patient satisfaction, investments in nurse well-being should be acknowledged as a strategic priority, in line with Saudi Arabia’s Vision 2030 objectives.

This research enhances the existing literature on QNWL, self-efficacy, and CF. Nonetheless, additional research is required to investigate the enduring impact of self-efficacy interventions on nurses’ performance and patient care outcomes. Future research should investigate the impact of organizational factors, including leadership style and institutional support, on nurses’ self-efficacy and work engagement. Furthermore, qualitative research may yield profound insights into nurses’ lived experiences concerning their QNWL and emotional well-being.

### 4.5. Study Limitations

The samples were drawn from two regions: Dammam and Riyadh. Therefore, the results are specific to these centers’ staff and cannot be applied to other Saudi Arabian primary care nurses. Second, the study used self-reported measures that may have been subject to recall bias and social desirability bias. Furthermore, this study assessed work life burnout, CF, and compassion satisfaction. Answers may vary over time based on the respondents’ personal and professional circumstances. The cross-sectional design of the study precludes conclusions about causality and does not allow for control of potential confounding variables. Instead, it reveals correlations between self-efficacy, CS, and QWL; research is required to confirm this hypothesis. Additional research employing diverse methodologies is needed to expand knowledge in this field, including future longitudinal and mixed-methods studies to confirm and build upon these findings. Further, there is a need for research that employs diverse methodologies to expand extant knowledge in this field. Finally, social and cultural dynamics unique to the Saudi healthcare environment, including family obligations and community perceptions of nursing, may have affected our results. The results may differ depending on the cultural attitudes and healthcare systems in other nations. Recognizing these contextual factors is essential for appropriately interpreting the results and for developing culturally sensitive interventions.

## 5. Conclusions

This study highlights the important connections between Saudi Arabian primary healthcare nurses’ self-efficacy, CS, CF, and QWL The results highlight the importance of fostering self-efficacy and reducing compassion fatigue to improve nurses’ work life quality. These findings can inform Saudi researchers and stakeholders to develop tailored workforce policies, supportive leadership strategies, and occupational health programs that can strengthen nurses’ well-being and care delivery. While context-specific, these insights may also benefit other healthcare systems with similar cultural and social dynamics.

## Figures and Tables

**Table 1 healthcare-13-01811-t001:** Demographic Characteristics.

Variable	*N*	%
Age		
22–24	7	4.4%
25–30	32	20.3%
31–45	101	63.9%
46–50	16	10.1%
Above 50	2	1.3%
Gender		
Male	50	31.6%
Female	108	68.4%
Educational Levels		
Diploma	47	29.7%
BSN	68	43.1%
MSN or higher	43	27.2%
Income		
SAR 6000 and less	11	7%
SAR 7000–9000	38	24%
Above SAR 9000	109	69%
Work Experience (Years)		
Up to 4 years	22	13.9%
5–9 years	36	22.8%
Above 9 years	100	63.3%
Overall Health Status		
Fair	2	1.2%
Good	20	12.7%
Very Good	48	30.4%
Excellent	88	55.7%

**Table 2 healthcare-13-01811-t002:** Mean Scores of the Main Study Variables.

Scales	M	SD	Interpretation
The General Self-Efficacy Scale	3.69	0.74	Moderate Levels of Self-Efficacy
Compassion Satisfaction	3.58	0.73	Moderate Levels of CS
Compassion Fatigue	2.73	0.50	Moderate Levels of CF
Overall Work-related QOL	3.26	0.52	Moderate Levels of work-related to QOL

**Table 3 healthcare-13-01811-t003:** Subscales of Overall WRQoL-2.

Variable	Mean ± SD	Description
Overall WRQoL-2	3.26 ± 0.52	Moderate work-related QOL
JCS	3.65 ± 0.67	A moderate scoring factor reflecting positive career satisfaction
SAW	2.46 ± 0.52	Moderate levels of SAW
CAW	3.50 ± 0.79	Moderate sense of control over work responsibilities
EEG	3.55 ± 0.85	Moderate levels of engagement and motivation
GWB	3.51 ± 0.60	Moderate levels of GWB
HWI	3.50 ± 0.79	Moderate balance between home and work life
WCS	2.60 ± 0.61	Moderate sense of dissatisfaction with work conditions

**Table 4 healthcare-13-01811-t004:** Mean Score Differences according to Individual Characteristics (*N* = 158).

Variables	GSE		CS		CF		WRQoL-2	
	Mean ± SD	*p*-Value	Mean ± SD	*p*-Value	Mean ± SD	*p*-Value	Mean ± SD	*p*-Value
Age								
22−30	30.44 ± 5.17	0.27	37.59 ± 7.11	0.08	52.61 ± 11.30	0.15	23.34 ± 3.53	0.32
31 and older	29.23 ± 6.22		35.23 ± 7.37		55.27 ± 9.76		22.67 ± 3.71	
Gender								
Male	30.22 ± 5.53	0.32	35.14 ± 6.52	0.43	53.65 ± 10.95	0.07	22.88 ± 3.88	0.91
Female	29.20 ± 6.19		36.12 ± 7.72		56.72 ± 8.03		22.82 ± 3.58	
Work Experience								
9 years or less	30.24 ± 5.90	0.25	37.26 ± 7.26	0.059	52.62 ± 11.17	0.06	23.70 ± 3.38	0.02
Above 9 years	29.79 ± 5.96		34.97 ± 7.32		55.78 ± 9.44		22.34 ± 3.75	
Income								
SAR 9000 or less	28.94 ± 6.06	0.41	33.88 ± 7.45	0.02	55.92 ± 9.72	0.28	23.42 ± 3.72	0.18
Above SAR 9000	29.79 ± 5.96		36.68 ± 7.18		54.04 ±10.39		22.57 ± 3.63	
Education								
Diploma	28.64 ± 5.78	0.18	34.60 ± 8.20	0.07	57.91 ± 9.49	0.004	21.67 ± 3.66	0.013
BSN	48.79 ± 5.98		35.32 ± 6.65		54.72 ± 9.23		22.95 ± 3.75	
MSN or higher	48.79 ± 5.98		37.91 ± 7.20		50.86 ± 11.27		23.93 ± 3.22	

**Table 5 healthcare-13-01811-t005:** Correlation Matrix for the Main Study Variables (*N* = 158).

Variable	1	2	3	4
The General Self-Efficacy	1			
2. CS	0.533 ***	1		
3. CF	−0.085	−0.124	1	
4. The work-related QOL	0.25 ***	0.233 **	−0.260 ***	1

Pearson’s correlation was used. ** *p* < 0.01; *** *p* < 0.001.

## Data Availability

The data presented in this study are available on request from the corresponding author due to privacy reasons.

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
