# Peer review of "Quality of Nursing Work Life, Compassion Fatigue, and Self-Efficacy Among Primary Care Nurses in Saudi Arabia"

_healthcare, 2025, doi:10.3390/healthcare13151811_

Round 1

Reviewer 1 Report

Comments and Suggestions for Authors

I would like to sincerely thank the authors for the opportunity to review this manuscript and commend them for addressing such an important topic in the field of nursing and healthcare. I appreciate the trust placed in the peer review process, and I hope that the following comments contribute constructively to enhancing the scientific quality and clarity of the article.

The introduction provides an appropriate context on quality of working life (QWL), compassion fatigue (CF), and self-efficacy in nursing practice. Relevant studies are cited, and the rationale for conducting the study in Saudi Arabia is supported.
However, although the conceptual framework is well outlined, it is recommended to better integrate the connection between the three main constructs (QNWL, CF, and self-efficacy) from the outset.

Additionally, more recent references (from 2023–2024) could be incorporated to update the contextual discussion. The rationale for selecting the study population could also be further developed, particularly from a public health or healthcare system perspective in Saudi Arabia.

The research questions are clear, relevant, and aligned with the purpose of the study.
However, the manuscript does not explicitly state the research objectives.
It is suggested to add a dedicated “Objectives” section to improve the structure and clarity of the research purpose.

Methodology: The study design is appropriate (quantitative cross-sectional). The population, sampling, sample size, instruments, and statistical analyses are clearly described.
Although G*Power was used, the manuscript does not present the formula or the specific parameters used for the power analysis (e.g., power = 0.80, n = 158).
The actual response rate and potential biases associated with non-probabilistic sampling are not discussed.

Instrument reliability and validity are well supported (Cronbach’s α > 0.70), but additional information is needed regarding the control of potential confounding variables.

The results are organized by construct, and the tables are clear. Mean scores, standard deviations, t-tests, ANOVAs, and Pearson’s correlations are presented.
However, a comprehensive table consolidating inferential results by sociodemographic group would be valuable.

Exact p-values and confidence intervals should be reported, and some descriptive data appear to be repeated both in the text and tables and should be revised.

The discussion is extensive and well structured. It effectively links findings with previous studies.
Nevertheless, key ideas could be synthesized to avoid redundancy. A more critical discussion of methodological limitations—such as social desirability bias and the constraints of a cross-sectional design—is needed.

Additionally, some paragraphs repeat information from the results section and should be reviewed for conciseness.

The conclusions are consistent with the findings but could be more concise.
It is recommended to strengthen the connection between the results and concrete practical implications (e.g., workforce policies, occupational health interventions) and include a brief reflection on the applicability of findings in other cultural contexts.

Tables should be numbered sequentially in the order they appear in the text. This would improve clarity and reference.

English Language and Style:

The manuscript is understandable; however, several redundant expressions and minor issues with style and cohesion are observed.

Examples:

  • This study examined the variation in these factors in relation to individual factors, including gender, education, income, and years of experience."
    Suggested: "This study examined how these factors varied according to gender, education, income, and years of experience.
  • Healthcare organizations should prioritize strategies that bolster nurses' confidence in their competencies.
    Suggested: Healthcare organizations should prioritize strategies to strengthen nurses’ professional confidence.
  • Accordingly, compassion encompasses attributes such as empathy, kindness, patience, and the ability to provide comfort.
    Suggested: Compassion includes attributes such as empathy, kindness, patience, and the ability to provide comfort.
    (Accordingly" is unnecessary and weakens the sentence’s clarity.)

Thank you once again for the opportunity to review your manuscript. I would like to congratulate the authors on their work and encourage them to consider the suggestions provided to further strengthen their research. I hope these recommendations will be helpful in improving the manuscript and supporting its successful publication.

Comments on the Quality of English Language

the manuscript is generally understandable, a thorough English language review by a native speaker is strongly recommended. Improving clarity, cohesion, and terminological precision.

Author Response

Greetings, 

See the attachment. 

Reviewer 2 Report

Comments and Suggestions for Authors

1 The abstract is correctly prepared, containing all the required information
2. The Introduction presents complete and well-organized information on the professional functioning of nurses, their role in the health care system, the importance of compassion and compassion fatigue and self-efficacy in their professional work, and the quality of professional life. It would be worthwhile to supplement the introduction with an indication/description of the specifics of nurses' work in Saudi Arabia - especially since there is little data in the literature on this subject. 
3 The description of the study sample lacks information on when the research was conducted. It would also be good to add information on whether the study sample was representative of this professional group (e.g., in terms of gender, age). 
4 The tools used are described properly.
5. The results are well organized and correctly presented.
6. The discussion is detailed, elaborate and includes references to the findings of other researchers. The applicability of the obtained results to the practice of health care and improvement of the quality of professional life of nurses is valuable. 
7. In the limitations of the study, the authors point out “social and cultural dynamics 
unique to the Saudi healthcare environment” - it would be worth pointing these out in more detail in the Introduction and the discussion of the results to show the cultural and social context of functioning in the professional role described.

Author Response

Greetings, 

See the attachment.

Reviewer 3 Report

Comments and Suggestions for Authors

Dear Authors,

       Congratulations on addressing such an important topic as  the relationships among self-efficacy, compassion fatigue, and the quality of nursing work life (QNWL) in primary care nurses, as the work of nurses plays a decisive role in the quality of care provided to patients.

         Kindly find the following feedback:

Line 107- The following four research questions guided this study – Are three or four research questions?

Line 116 - Materials and Methods

Please kindly specify:

- the study period

- the number of participants to whom the questionnaire was sent

-  the number of nurses who were excluded

- the response rate

Line 480-482: The limitations of this study highlight the necessity for additional research that employs a variety of methodologies to expand the knowledge in this field - we suggest that you move it within the paragraph 4.5 Limitations of the Study

Author Response

Greetings, 

See the attachment.
